# Using an IT-Based Algorithm for Health Promotion in Temporary Settlements to Improve Migrant and Refugee Health

**DOI:** 10.3390/healthcare9101284

**Published:** 2021-09-28

**Authors:** Elena Riza, Achilleas Lazarou, Pania Karnaki, Dina Zota, Margarita Nassi, Maria Kantzanou, Athena Linos

**Affiliations:** 1Department of Hygiene, Epidemiology & Medical Statistics, Medical School National and Kapodistrian University of Athens, 11527 Athens, Greece; achilles.lazarou@gmail.com (A.L.); margrh@med.uoa.gr (M.N.); mkatzan@med.uoa.gr (M.K.); 2Prolepsis Institute for Preventive Medicine and Environmental and Occupational Health, 15121 Marousi, Greece; p.karnaki@prolepsis.gr (P.K.); d.zota@prolepsis.gr (D.Z.); a.linos@prolepsis.gr (A.L.)

**Keywords:** information technology, migrants, refugees, health promotion, health literacy, temporary settlements, obesity, vaccinations, dental care

## Abstract

The application of the electronic algorithm developed by the Mig-Healthcare project was pilot tested in a sample of migrants and refugees in 2 Reception and Identification Centres (RICs), temporary settlements, in Greece using portable devices. The questions relate to health literacy issues, to mental health, to vaccination history, to lifestyle habits such as smoking, alcohol intake, diet, to the presence of diseases such as heart disease or diabetes, to the use of prevention services and to dental care. A total of 82 adults, 50 women and 32 men, participated. Data analysis showed that 67.1% (55) of the respondents had difficulty in understanding medical information and 57.3% (47) did not know where to seek medical help for a specific health problem. Four main areas of health problems were identified and further action is required: (A) mental health concerns, (B) vaccinations, (C) obesity, and (D) dental hygiene. Direct linkage with the “Roadmap and Toolbox” section of the project’s website gave the respondents access to many sources and tools, while through the use of the interactive map, specific referral points of healthcare delivery in their area were identified. IT-based intervention in migrant and refugee populations in Greece are effective in increasing health literacy levels and identifying areas for health promotion interventions in these groups. Through linkage with the project’s database, access to healthcare provision points and action to seek appropriate healthcare when necessary are encouraged. Given the attenuated vulnerability profile of people living in temporary settlements, this algorithm can be easily used in primary care settings to improve migrant and refugee health.

## 1. Introduction

Migration has been present in the world as people have always tried to secure favourable and safe living conditions. Since 2015, however, there has been a great influx of migrants and refugees into the countries of Europe, mainly as the result of poverty, under-development, and civil war in countries of Africa and of the Middle East. According to data from the International Organisation of Migration (IOM) and the United Nations High Commissioner for Refugees (UNHCR), over 1.5 million migrants and refugees have arrived in Europe through the sea route, but also by land in search of a better future [1].

According to the IOM, the UN-related intergovernmental organization for the support of migrants and refugees, migration is defined as “the movement of a person or a group of people, either across an international border, or within a state. It is a population movement, encompassing any kind of movement of people, whatever its length, composition and causes; it includes the migration of refugees, displaced persons, economic migrants, and persons moving for other purposes, including family reunification” [2].

The journey of migration from their home countries to Europe, whether voluntary or forced, is usually long and very dangerous and it is often related with complex experiences and conditions that reshape people’s lives [3]. Moreover, it exposes migrants to several health risks (e.g., extreme weather conditions, exploitation, physical and mental trauma) that may create health issues through accidents, fear, or extreme distress and may worsen pre-existing health conditions such as diabetes and cancer.

The increasing number of migrants in European countries impacts the respective healthcare systems as they face new and immediate healthcare needs in populations with different characteristics and cultural backgrounds. Moreover, the healthcare systems have to adapt and prepare to accommodate the needs of these people in the future. Research has shown that the most prevalent healthcare needs in newly arrived refugees and migrants are at the primary care level and are most needed in the first months after arriving in the host country [4]. These needs often refer to mild respiratory issues, musculoskeletal and digestive problems, pregnancy checks and reproductive care [5,6,7].

In this respect, Mig-Healthcare (www.mighealthcare.eu), a 3-year project funded by the European Commission, aimed to define the elements of best practices to help the health of migrants and refugees at the community level and to develop tools that can assist in this process [8]. One of the deliverables of this project is an electronic algorithm in the form of a questionnaire that has been designed to help increase the health literacy level of migrants, to assist health professionals in prioritizing areas for health promotion intervention and to identify health issues for which a medical consultation is required.

The level of health literacy is a very important factor in the prevention and the effective management of diseases and it also affects the level of control a person has over making informed decisions about their own health [9]. People with low health literacy levels tend to seek emergency treatment and require hospitalisation more frequently than others and also show higher levels of morbidity and mortality [10].

Information Technology (IT)-based approaches have played a significant role in delivering health promotion programmes and in improving health literacy in vulnerable population groups [11,12]. Health promotion as a key activity to improve health, especially at the primary and secondary levels, entails an element of education and training. As such, health literacy, which is defined as “the degree to which individuals have the capacity to obtain, process and understand basic health information and services needed to make appropriate health decisions” [13], becomes crucial in ensuring the motivation and competence of the target population groups to access and to use health information and practices. Many people nowadays are aware of and understand health information to such a degree where they are able to make suitable decisions with regards to their health. However, there remains a large population of people who do not know or do not have access to such health information, which means that they have poor health literacy and are unable to make such decisions. A notable group of people are refugees and migrants, who, in addition to the perils of the migration journey, face additional barriers in using and accessing health care, namely, cultural, linguistic, and practical barriers in the host countries.

In this study, we used the electronic questionnaire developed by the Mig-Healthcare project to pilot test its application in a sample of migrants and refugees residing in two Reception and Identification Centres (RICs), temporary settlements, in the Attica area of Greece. There has been a high influx of migrants and refugees in the country since 2015, with the majority arriving in the country via the Mediterranean sea route on the North Aegean Islands or on the Greek mainland residing in state-organised settlements for a certain period of time (ranging from a few months up to several years) while waiting to sort out their legal status. A significant proportion of them will remain in Greece while others will move to other destinations, mostly within Europe. This high influx of refugees in the country has put a tremendous strain on the Greek National Health Service, which has stressed the importance of strengthening the system by expanding the primary care network and by finding alternative ways to best meet the needs of newly arrived refugees and migrants [14,15]. Our effort aimed at testing the potential of using the electronic algorithm in low-resource primary care settings with the help of health professionals in the settlements to help improve the health status of migrants and refugees, to increase health literacy, and to facilitate their integration in the host communities.

## 2. Materials and Methods

The electronic algorithm was used to gather primary data regarding the health of the migrants and refugees residing in RICs, temporary settlements in Greece. The questionnaire was completed in the medical unit of each camp with the help of a health professional and of an interpreter to ensure proper communication. The questionnaire for this pilot phase was delivered in the English language, but availability in some common languages of migrants and refugees is foreseen. Upon completion of the questionnaire, a report can be downloaded stating the health issues in which further action is required and guidance is given as to where to look for help (healthcare system navigation information available on the project’s site). The questionnaire is freely available online through the project’s platform and it can be completed in 5–10 min depending mainly on the digital ability of the person and on the quality of the internet connection. The entire process is done completely anonymously since no identifiable personal data are required.

The questionnaire consists of a set of 25 questions which relate to health literacy issues, to mental health, to vaccination history, to lifestyle habits such as smoking, alcohol intake, diet, to the presence of diseases such as heart disease or diabetes, to the use of prevention services, and to dental care.

With the use of portable electronic devices (tablets with Wi-Fi access), migrants and refugees were asked to complete the questionnaire with the assistance of an interpreter in two Reception and Identification Centres (RIC) in the Attica area during April–May 2020. Migrants and refugees residing in these centres are people who have been in the country from a few months to a few years, while they are waiting for their legal status to be settled. Studies have shown that, although they are generally healthy at least when they arrive to the host country (the healthy migrant effect), they are vulnerable groups of people who frequently have limited access to health care.

A total of 82 adults completed the questionnaire anonymously during April–May 2020, with the use of tablet devices that had access to the electronic tool on the Mig-Healthcare project’s platform. Internet access was secured with the provision of adequate amounts of data (charge cards were given). The tablets were available in the medical unit of each RIC. The forms are saved on the project’s drive, where they can only be accessed by authorised researchers. The algorithm involved with the questionnaire is able to then calculate areas which require further attention based on the responses.

The majority of residents in these two RICs come from Syria and Afghanistan, but there are also people from Somalia and Nigeria. Due to the fact that the study took place during the COVID-19 pandemic, several restriction measures were in force in the settlements which did not allow us to apply a random selection process for participation in the study, as we were not allowed to freely move inside the camp. Therefore, we relied on participation from refugees who had a reason to visit the camp’s health unit. Moreover, due to the restrictions, the special permit required from the authorities to access the camp was very difficult to obtain. As a result, the communication between the study research team and the healthcare workers inside the camps was very limited and it was not possible to organise open information sessions to inform the residents on the study. As such, the best option was the purposive sampling method, which resulted in a relatively low number of study participants. For data protection reasons, neither age nor nationality was recorded in the report forms. Verbal informed consent was requested from the participants prior to the completion of the anonymous questionnaire by the health professional working in each RIC, after explaining the scope of the pilot action, the benefits for the participants and requested their agreement to be informed on the outcomes of the algorithm. The study protocol was approved by the Ethics Committee of the Medical School, National and Kapodistrian University of Athens.

## 3. Results

A total of 82 adults completed the electronic questionnaire; 50 women and 32 men. In questions regarding access to healthcare services, data analysis shows that there was difficulty from 67.1% (55) of the respondents in understanding medical information and 57.3% (47) answered that they did not know where to seek medical help for a specific health problem.

### 3.1. Identifying Health Issues

From the responses in the sections regarding health issues, four main areas of health problems were identified, where further action is required for several of the respondents. These include: (A) mental health concerns, (B) vaccinations, (C) obesity, and (D) dental hygiene.

#### 3.1.1. Mental Health

The algorithm uses the questions on mental health based on the short version of the Medical Outcomes Study: SF-36 [16], which is standardised for use in vulnerable populations such as migrants and refugees. The questions refer to how the person has felt the past four weeks and they are ranked on a six-point scale, where 1 signifies the response “all the time” and 6 represents “none of the time.” In each question, each answer corresponds to a set numerical value in the range of 0 to 100; however, the high score in each question does not always mean a better health state. Based on a scoring key given by the researchers Medical Outcomes Study [16], an overall score out of 100 is reached, which corresponds to a health state. Scores under 50 are considered indicative of a poor mental health state.

The study’s algorithm calculated overall score (out of 100 based on the scoring key of all the answers given), is then used to indicate the need to consult a specialist to assess the particular issue. The calculation was based on a series of questions which were formatted as shown below.
Q: How much of the time during the past 4 weeks…?(1) All of the time (2) Most of the time (3) A good bit of the time (4) Some of the time (5) A little of the time (6) None of the time1. Have you been a very nervous person?1234562. Have you felt so down in the dumps 123456that nothing could cheer you up?3. Have you felt calm and peaceful? 1234564. Have you felt downhearted and blue?1234565. Have you been a happy person? 123456

In the sample of the study, 38 out of 82 people (46.3%) scored a value less than 50, which corresponds to poor mental health or poor emotional well-being, and were subsequently advised to consult a specialist. Moreover, in the “Roadmap and Toolbox” section of the Mig-Healthcare project’s site, a drop-down list was shown upon completion of the section with relevant information in various forms to choose from. A message to use the interactive map also available online appeared in order to help locate the nearest healthcare service point. This not only helped the participants on the identified mental health issues, but it also provided information about emotional well-being, key signs of issues and ways in which the participant can deal with issues by themselves or by talking to a specialist. Ultimately, these resources, with guidance from the health professional of the RIC, can lead to an increased awareness of mental health issues and improve health literacy about their own mental health.

#### 3.1.2. Vaccinations

Vaccinations, especially for infectious diseases, are particularly important for people who live in temporary settlements or camps, since there is high potential for transmission for diseases such as measles or tetanus due to the overcrowded and unhygienic conditions. The participants in the sample were all adults and they were asked about their own vaccination history.

It was interesting to see that 47 out of the total of 82 people (57.3%) reported that they were either never vaccinated or they were only vaccinated as children (Table 1), which means that they were not protected against some diseases that require booster shots such as tetanus. People in this situation received a message to seek further advice on the required vaccinations according to the national vaccination guidelines. Through the Mig-Healthcare website, they had access to tools in various languages to find out more information on the use of vaccines, protection from diseases and common vaccination practices.

#### 3.1.3. Obesity

Obesity is a growing problem in many societies, especially in the Western part of the world, and there are indications that migrants and refugees tend to adopt the dietary habits of the host country [17], which means that obesity is a concern for these people as most of them are planning to stay in the European countries. Obesity is a major risk factor for the development of cardiovascular disease and of type 2 diabetes, diseases that make up the greatest part of disease burden globally [18].

The algorithm used in this study has a function to calculate Body Mass Index (BMI) as an indicator to show the level of obesity of the individual based on the values of weight (kg) over height squared (m^2^). In this application, the BMI is calculated based on self-reported values of height and weight, but they can also be measured by a health professional if required.

The categories of BMI as indicated by the World Health Organization (https://www.euro.who.int/en/health-topics/disease-prevention/nutrition/a-healthy-lifestyle/body-mass-index-bmi, accessed on 19 September 2021) are as follows:

Underweight < 18.5 kg/m^2^

Normal weight 18.5–24.9 kg/m^2^

Overweight 25–29.9 kg/m^2^

Obese > 30 kg/m^2^

According to the responses, 12 people (10 women, 2 men), or 14.6% of the sample, had a BMI of over 30 kg/m^2^ and were classified as obese (Table 2). Following the findings, information and many tools about the effects and ways of combatting obesity were at their disposal through the project’s database. The additional information presented included nutrition information, ways to maintain a healthy diet and, as in the previous steps, they could use the interactive map to find nutrition specialist services in their area for further help.

#### 3.1.4. Dental Hygiene

Dental hygiene is often overlooked by people on the move such as migrants and refugees [19] because of the difficult living conditions that in some cases include limited access to clean water and the consumption of high-sugar foods. Lack of proper dental care is also a problem, because, in most countries, there is no system in place to cover the costs of dental treatments, which are generally quite high. Therefore, many people, and children in particular, do not take care of their teeth, but only visit the dentist once an issue arises (most frequently tooth pain or gum bleeding).

As shown in Table 3, for the question “When did you last visit a dentist?”, 23 people out of the 82 who completed the questionnaire replied with “never”, indicating the need for intervention and provision of dental care.

On the Mig-Healthcare website (Figure 1), there is a lot of information available regarding dental care and the steps required to maintain good oral hygiene in general. Included in this information are ways to distinguish different dental issues, which have the potential to lead to quicker and more efficient treatment as well as potential prevention of further damage to the teeth and the oral cavity.

## 4. Discussion

The application of an electronic algorithm in a sample of migrants and refugees living in reception and identification camps in Greece has helped to identify gaps in the understanding of health concepts such as understanding medical information in leaflets, and it has provided them with useful links to tools to increase their knowledge in several thematic areas. Moreover, through the use of the interactive map, the study participants had the opportunity to locate points of care that they can access to seek professional help for a medical issue.

It is also very important to note that, through the use of the algorithm, the participating migrants and refugees were able to perform a quick self-health assessment and to point out areas where help from a health professional is required, such as mental health support, advice on vaccination, body weight control and dental care. These areas have also been identified through research as important in groups of vulnerable people such as the ones in our sample [11,20].

The areas of intervention identified through the use of the algorithm can lead to an increase in the health literacy of the users by providing them with information in various forms through the “Roadmap and Toolbox” section that is available on the project’s database in different forms and in different languages.

A generalised use of this electronic algorithm has the potential to act as a health education tool for migrants and refugees which, if combined with access to healthcare services, can lead to the design of health interventions by the health authorities that consider the specific needs and the characteristics of the target population.

Migrants and refugees often have a low level of heath literacy which, in many cases, is combined with communication difficulties in the host country’s language, different cultural backgrounds, perceptions, and attitudes towards health. These factors often lead to the migrants finding themselves without help, outside healthcare systems. Studies in countries such as Germany and Sweden have shown that over 70% of migrants report difficulties in understanding health-related information [21]. So, by improving the level of health literacy, the health status of these people will also improve, which will have a positive impact on their well-being and on the public health level of the host society.

Moreover, migrants and refugees are classified as vulnerable groups with characteristics that make them susceptible to worse health outcomes. In times of difficulties, such as the 2011 global economy crisis or the current COVID-19 pandemic, social inequalities become exacerbated and adversely affect these groups [22]. Certainly, along with migration, other factors such as gender, ethnicity and level of education are predisposing factors of health and social inequalities.

Being a migrant or refugee is a factor that socially determines health and must be considered in the efforts to reduce health inequalities. Such efforts mainly address the state of health and the ease of access to the health services [23]. The state of one’s health can be improved by actions to secure better housing, quality food, education, language support, knowledge on diseases, among others. Access to health can be increased by actions to remove any language barriers, to provide culturally appropriate health-related information and to facilitate the identification of available health services.

Attention should be given in order to ensure that all efforts observe the ethical principles of respect and justice. The use of IT solutions in minimising health inequalities is essential and offers many advantages in securing the ethical prerequisites. Under the principle of respect lies autonomy and protection of those who are vulnerable. IT applications in health care for migrants and refugees enhance their autonomy by giving them information that can help them make informed choices for their own health or for the health of family members. In addition, their level of health literacy increases, which enables them to make future decisions without much external help, potentially preventing certain health issues. Digital applications have the potential to observe the principle of justice as they help remove several barriers in the delivery of health care.

The applications of computer technology are available online and take advantage of the fact that most people have a device readily available that they can use. Digital applications can help improve health literacy, can promote healthy behaviour, and can help to manage diseases. Moreover, once they are set up, they are relatively cheap to maintain and do not require storage space [24]. Technology has the potential to lead to better health outcomes, especially in vulnerable people such as migrants and refugees through the use of computers and mobile devices by monitoring specific disease indicators such as weight, heart rate or by gaining access to several information sources with information on health behaviour and disease prevention. These applications may prompt people to seek help on a health issue and act upon it early. The methods used traditionally such as the leaflets and brochures may no longer be as effective since they have a high production cost and they can reach fewer people compared to electronic information.

The report “European citizens’ digital health literacy” from the European Union indicates that digital applications for health can reduce social inequalities and encourages their use in an effort to increase equal access to health care for all [25]. These digital heath tools can improve the understanding of health-related information and will allow them to make better use of health services, also considering factors such as age, gender, nationality, level of education and time in the host country.

### 4.1. Limitations

The limitations in the present study mainly arise from the fact that refugees and migrants in camps are an inherently hard-to-reach population with different attitudes and beliefs towards health and health prevention in particular and their participation in health-related activities is often limited. Moreover, the time period during which the study was conducted was severely affected by the COVID-19 pandemic and the imposed measures, which hindered our access to the camps and limited the size of the study participants. It must be noted that, in order to access the camps, special permission by the regulating authorities was required. As the data collection took place in the health unit inside each RIC, freedom of movement around the camp was restricted, so, in our study sample, we were only able to include people who visited the health unit during specific allocated days and times. Given these conditions, our intention to organise open information sessions to inform the residents about the study could not be implemented. As a result, it was not possible to apply a random selection process for participation, nor to conduct a between-group comparison (algorithm users vs. non-users) to further analyse the effect of the intervention. Due to the need of an interpreter and the limited technological ability of the respondents in combination with their low health literacy level, the data collection process was much slower than anticipated, further limiting the number of people who were able to complete the algorithm and to navigate the project’s database. Moreover, since our main aim was to pilot test the application of the algorithm in a temporary, emergency health setting, we did not collect detailed personal identifiable information from the participants, which did not allow us to examine health issues in greater detail and to present high-risk disease profiles in the study population.

### 4.2. Further Research

According to the WHO report on migrant health in the European region [26], migrants and refugees tend to adopt the health profile of the population in the host countries. As such, it is expected that in the coming years, the cases of non-communicable diseases and mental conditions will increase. Many of these conditions, such as cardiovascular disease, some forms of cancer, and respiratory conditions, are largely preventable through health education, health promotion, early detection and self-management. Digital technology applications, such as the Mig-Healthcare algorithm, are user-friendly and easily accessible. They can be used in most settings and they have the potential of collecting data that can be used to fill the existing information gaps in migrant and refugee health. Our study has indicated the potential of a digital application in increasing the health literacy level and the health awareness in a sample of a vulnerable population. Further interventions are required to demonstrate the effectiveness of such tools at the local community level and into the primary healthcare system, where tailor-made activities can be designed to target the needs of each group. These can also be facilitated by including additional personal detailed demographic and medical data into the algorithm. Such interventions can best operate under official agreements between the authorities in charge of the temporary settlements (usually the state migration services) and the research groups (e.g., academia, NGOs), in order to ensure increased participation, long-term follow-up, confidentiality, wider operationalisation of the interventions and linkage with the local primary care network. In this way, the effectiveness per population group, the strong and the weak points of the intervention can be identified and changes can be made accordingly. This approach contributes to the achievement of the UN’s Sustainable Development Goals “Good health and well-being” (Goal 3) and is in line with the Health in All policies approach and the WHO “Universal health coverage-leave no one behind” commitment [27].

## 5. Conclusions

This study has helped to identify the effectiveness of an IT-based intervention in refugee and migrant populations in Greece in increasing health literacy levels and identifying areas for health promotion interventions in these groups. Our results are in line with those from several other studies in identifying health issues of concern, such as poor mental health, need for vaccination, obesity and poor dental hygiene [21]. Through linkage with the project’s database, it also indicated access to healthcare provision points and prompted action to seek appropriate health care when necessary. Given the attenuated vulnerability profile of people living in temporary settlements, this algorithm can be easily used in primary care settings to improve migrant and refugee health in a cost-effective way.

## Figures and Tables

**Figure 1 healthcare-09-01284-f001:**
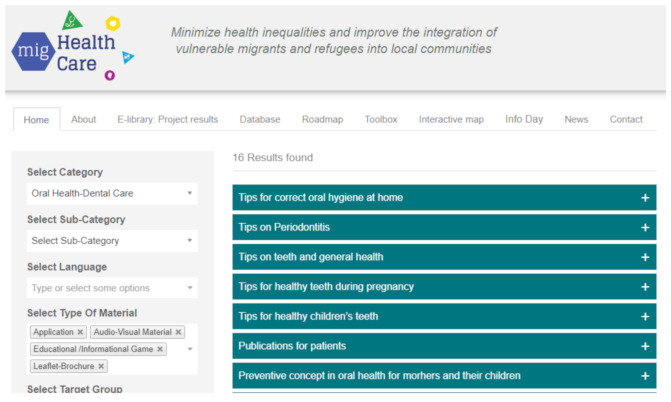
Screenshot of the tools and information regarding dental care.

**Table 1 healthcare-09-01284-t001:** Question 12: “When was the last time you were vaccinated?”

Gender		Less than 10 Years	Never/As a Child	Total
Female	Count	27	23	50
	% of Total	32.9%	28%	60.9%
Male	Count	8	24	32
	% of Total	9.8%	29.3%	39.1%
Total	Count	35	47	82
	% of Total	42.7%	**57.3%**	100.0%

Bold, the very high amount of unvaccinated persons.

**Table 2 healthcare-09-01284-t002:** BMI categorization of the study participants.

BMI Categories	Underweight	Normal Weight	Overweight	Obese	Total
Females	1	24	14	10	50
Males	0	17	14	2	32
Total	1	41	28	12	82

**Table 3 healthcare-09-01284-t003:** Question 22 “When did you last visit a dentist?”

Gender		Never	The Last 2 Years	Total
Female	Count	11	39	50
	% of total	13.4%	47.6%	61%
Male	Count	12	20	32
	% of total	14.6%	24.4%	39%
Total	Count	23	59	82
	% of total	**28%**	72%	100.0%

Bold, the high amount of people wha have NEVER visited a dentist.

## Data Availability

The data collected for this study contain no identifiable information, are stored by the principal researcher (E.R.) and are available upon request.

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
