# Peer review of "Using an IT-Based Algorithm for Health Promotion in Temporary Settlements to Improve Migrant and Refugee Health"

_healthcare, 2021, doi:10.3390/healthcare9101284_

Round 1

Reviewer 1 Report

I only suggest to the Authors to include in the introduction some comments about other relevant references.

For instance, IJERPH between 2017 and 2019 published a special issue dealing with "Refugee, migrant and ethnic minority health", which included 38 very interesting and original papers that it is worth mentioning.

Author Response

Dear Reviewer,

thank you very much for your valuable comments. Please find attached a pdf file with the changes in our manuscript.

Kind regards

Reviewer 2 Report

This is an interesting and descriptive study of a timely matter relevant to improve the healthcare and wellbeing of migrants and refugees.  A small number of adult subjects was selected (N=82). The analysis of limited data reveals the importance of reaching out to vulnerable populations. The paper has to make specific amendments as follows:

  1. Describe how the sampling procedure was executed and the limitation of the purposive sampling method.
  2. Explain the term "technology intervention" applied to the evaluation of help-seeking behaviors.
  3. Explain why RCT design was not considered in the analysis of intervention effect.
  4. Provide more details on measurement instruments and scales used.
  5. Document specific limitations of the study and explain how future research should be conducted.
  6. Add a few of the case reports enhancing the credibility of the study.
  7. Discuss the generalizability of the findings.

Author Response

(The authors gave the same response as above.)

Round 2

Reviewer 2 Report

The authors are responsive to the critique. Necessary statements are added to offer reasonable answers to the questions.